# Effect of Drought on Photosynthesis of Trees and Shrubs in Habitat Corridors

Josef Urban [1,2,*], Marie Matoušková [1], William Robb [3], Boleslav Jelínek [1] and Luboš Úradníček [1]

[1] Department of Forest Botany, Dendrology and Geobiocenology, Faculty of Forestry and Wood Technology, Mendel University in Brno, Zemědělská 3, 61300 Brno, Czech Republic; marie.matouskova@mendelu.cz (M.M.); boleslav.jelinek@mendelu.cz (B.J.); lubos.uradnicek@mendelu.cz (L.Ú.)

[2] School of Ecology and Geography, Siberian Federal University, 79 Svobodny pr., 660041 Krasnoyarsk, Russia

[3] Department of Engineering, Faculty of Forestry and Wood Technology, Mendel University in Brno, Zemědělská 3, 61300 Brno, Czech Republic; billyrobbb@gmail.com

[*] Correspondence: josef.urban@email.cz

**Abstract:** Drought and high evapotranspiration demands can jeopardise trees and shrubs in windbreaks and habitat corridors, where they are more exposed to the effects of extreme weather than in the forest. This study utilised chlorophyll fluorescence to assess how the leaf-level physiological processes of 13 woody species typically planted in Czech habitat corridors responded to the effects of naturally occurring drought and their ability to recover after rain. Linear electron flow (*LEF*) responded only weakly to the drought, indicating high levels of photorespiration. Trees and shrubs increased the proportion of energy which was dissipated in a harmless way ($\Phi_{NPQ}$) during drought and decreased the proportion of energy dissipated through non-regulated processes ($\Phi_{NO}$). In this way, they reduced processes potentially leading to the production of reactive oxygen species. All species except *Tilia cordata* Mill. maintained high $\Phi_{NPQ}$ even after its release from drought. *Tilia cordata* was potentially the most susceptible tree to drought due to its low *LEF* and high $\Phi_{NO}$. The most drought-resistant tree species appeared to be *Acer campestre* L. and shrubs such as *Prunus spinosa* L., *Viburnum lantana* L, and *Crataegus monogyna* L. These shrubs may be planted at the sunny edges of habitat corridors. The woody species identified as resistant to drought in habitat corridors may also be considered resistant in a warming climate or suitable for planting in the urban environment which is generally warmer and drier than in a forest.

**Keywords:** biocorridors; habitat corridors; drought; chlorophyll fluorescence; drought resistance; *Acer campestre* L.; *Carpinus betulus* L.; *Tilia cordata* Mill.

## 1. Introduction

Habitat corridors, shelterbelts, and biocorridors are narrow strips of woody vegetation in an otherwise urban or agricultural landscape. They connect fragmented habitats, allowing for the movement of plants and animals between them. They play a crucial role in maintaining biodiversity, facilitating gene flow, and enhancing ecosystem resilience [1]. The importance of habitat corridors has been widely recognised in conservation biology, as they can help mitigate the negative impacts of habitat fragmentation and climate change on species populations [2]. In recent years, scientific research has increasingly focused on the ecological and functional roles of habitat corridors, as well as on the factors that influence their effectiveness [3,4]. This paper focuses on the assessment of various tree species in the habitat corridors, especially with respect to drought.

Trees and shrubs in habitat corridors are exposed to multiple stressors. Drought is one of the most pronounced stress factors in shelterbelts and the main reason for their degradation [5]. Transpiration from the shelterbelts commonly exceeds the reference evapotranspiration [6]. While shelterbelts decrease the wind speed above the surrounding

crops [7,8], the trees and shrubs in the belt are exposed to solar radiation and turbulent wind [9]. Increased transpiration inevitably leads to a decline in the soil moisture within and around the shelterbelt [10,11], further imposing drought on the trees. During prolonged periods of drought, plants experience a decline in water availability, which leads to reduced photosynthesis and other physiological processes [12]. As a result, plant growth is stunted, and the plants may experience wilting, leaf shedding, and even death [13–15]. Previous studies have shown that drought can also cause changes in plant anatomy, morphology, and physiology such as reduced leaf area, increased root-to-shoot ratio, and decreased stomatal conductance [16]. Additionally, the severity of drought effects can vary among species and can depend on several factors, including the plant's water-use strategy, root depth, and growth rate [17]. Understanding the effects of drought on trees and shrubs is critical for predicting the impacts of climate change on habitat corridors and for developing effective strategies for mitigating these impacts [18].

The first response of the trees to drought is stomatal closure [19]. While the closed stomata reduce water vaporisation from the leaves, they also act as a resistance against the diffusion of $CO_2$ into the leaves. Since photosynthesis and stomatal conductance are tightly coupled [20], the level of water stress can be indirectly estimated from the response of photosynthesis [21]. Therefore, the response of photosynthesis to drought and its recovery after drought release may provide a clue on the resistance of different tree species [22]. In particular, chlorophyll fluorescence provides a rapid tool to quickly screen the drought response of multiple tree species in the habitat corridor [23]. Different tree species exhibit varying degrees of resistance to drought, and such resistance can be reflected in their chlorophyll fluorescence parameters [24,25]. For instance, drought-tolerant species generally exhibit higher values of maximum quantum efficiency of photosystem II ($F_v/F_m$) and lower values of non-photochemical quenching (*NPQ*), indicating better photosynthetic performance and lower susceptibility to photodamage under drought conditions [26]. In contrast, drought-sensitive species typically exhibit higher *NPQ* values, suggesting a decline in photosynthetic efficiency and an increase in energy dissipation to avoid photodamage. In addition, in drought-sensitive species, more energy will be dissipated in a nonregulated way, which may lead to the production of reactive oxygen species [27]. Therefore, chlorophyll fluorescence analysis can provide valuable insights into the mechanisms underlying the drought resistance or sensitivity of different tree species, which is crucial for predicting the impacts of drought on habitat corridors and developing effective management strategies.

This study aimed to assess the drought resistance of thirteen tree and shrub species commonly used in the habitat corridors in the Czech Republic. We compared the change in indices of chlorophyll fluorescence between the wet and dry periods and the ability to recover after the drought stress ceased. The dry summer of 2022 [28,29] provided a good opportunity to study the trees' response to drought in situ. Because the index $F_v/F_m$ measured on the dark-adapted leaf is rather insensitive to mild levels of drought [30,31], we rather focused on the indices measured on the foliage adapted to ambient light. Specifically, we focused on the linear electron flow (*LEF*), which is derived from the yield of photosystem II ($\Phi_{II}$). Two other parameters we focused on, $\Phi_{NPQ}$ and $\Phi_{NO}$, were related to the two ways of energy dissipation, by regulated and non-regulated processes, respectively [32]. We aimed to use these three indices to assess how the different tree species respond when soil moisture becomes limited. The second aim was to assess the ability of the thirteen woody species to recover after the release from drought stress. The experiment was conducted in the habitat corridors in the east of the Czech Republic, under naturally occurring drought in the summer of 2022.

## 2. Materials and Methods

### 2.1. The Study Site

The Křižanovice habitat corridor was established in the autumn of 1990 on arable land north of the village of Křižanovice u Vyškova (49°17′58.837″ N, 17°2′14.324″ E). This

habitat corridor is located in an area with a mean annual temperature of 9.0 °C and mean annual rainfall of 549.7 mm. According to the last pedological survey, there are Luvic Chernozems in the habitat corridor and Haplic Chernozems in its surroundings (arable land) [33]. The habitat corridor is formed by three segments with different azimuthal orientations. One permanent research plot was established in each of them and repeated dendrometric surveys were carried out there (Table 1). In the habitat corridor, *Acer campestre* L., *Acer platanoides* L., *Carpinus betulus* L., *Cornus alba* L., *Cornus mas* L., *Cornus sanguinea* L., *Crataegus monogyna* Jacq., *Fraxinus excelsior* L., *Ligustrum vulgare* L., *Prunus mahaleb* L., *Prunus spinosa* L., *Quercus robur* L., *Rhamnus cathartica* L., *Rosa canina* L., *Tilia cordata* Mill., and *Viburnum lantana* L. were planted. Except for *Cornus alba*, all species have survived to the present day. Additionally, *Sambucus nigra* L. has spread from the surroundings into the habitat corridor. All woody species in the habitat corridor were deciduous broadleaves, with the C3 type of photosynthesis.

**Table 1.** Dendrometrical characteristics of trees and shrubs in the habitat corridor. Height of tallest and shortest tree, mean tree height, breast height diameter of the thickest and thinnest tree, and mean diameter of a specific species in the habitat corridor.

| | Family | Character | Height (cm) | | | Breast Height Diameter (mm) | | |
|---|---|---|---|---|---|---|---|---|
| | | | **Min** | **Max** | **Mean** | **Min** | **Max** | **Mean** |
| *Acer campestre* | *Sapindaceae* | Tree | 80 | 1100 | 817 | 22 | 315 | 117 |
| *Acer platanoides* | *Sapindaceae* | Tree | 650 | 1500 | 1231 | 70 | 353 | 201 |
| *Carpinus betulus* | *Betulaceae* | Tree | 485 | 870 | 678 | 61 | 76 | 69 |
| *Crataegus monogyna* | *Rosaceae* | Shrub | 130 | 460 | 343 | | | |
| *Fraxinus excelsior* | *Oleaceae* | Tree | 290 | 1600 | 1166 | 51 | 325 | 154 |
| *Ligustrum vulgare* | *Oleaceae* | Shrub | 50 | 324 | 173 | | | |
| *Prunus mahaleb* | *Rosaceae* | Tree | 207 | 1950 | 767 | 16 | 95 | 49 |
| *Prunus spinosa* | *Rosaceae* | Shrub | 240 | 400 | 298 | | | |
| *Quercus robur* | *Fagaceae* | Tree | 1200 | 1250 | 1225 | 500 | 660 | 580 |
| *Rhamnus cathartica* | *Rhamnaceae* | Shrub | 287 | 555 | 405 | | | |
| *Sambucus nigra* | *Viburnaceae* | Shrub | 407 | 630 | 483 | | | |
| *Tilia cordata* | *Malvaceae* | Tree | 185 | 1550 | 942 | 19 | 318 | 149 |
| *Viburnum lantana* | *Viburnaceae* | Shrub | 105 | 407 | 255 | | | |

### 2.2. Methods

Air temperature, relative air humidity, global radiation, and precipitation were measured by a weather station (EMS Brno, Czech Republic) on an open plot above a grass area, less than 1 km from the studied habitat corridors (Figure 1). Vapor pressure deficit (*VPD*) and reference evapotranspiration ($ET_0$) were calculated using the FAO approach [34] when the wind speed was set to 2 m s$^{-1}$. Soil moisture was measured at a depth of 0–15 cm by a TMS-4 (Tomst, Czech Republic) in three replications.

Physiological measurements were performed on cloudless or almost cloudless days (14 June, 25 July, and 2 September 2022, Supplementary Figure S1) with different levels of drought stress. The dynamics of soil moisture during the year 2022 with arrows indicating the days of measurements are depicted in Figure 2. The soil water content of 21.3% on 25 July was close to the seasonal minimum of 21%, thus indicating that the soil was dry. Additional information from the Intersucho service [29] indicated that while June and September were relatively wet, July was extremely dry (Supplementary Figure S2). In the text, these three days are therefore respectively referred to as wet, dry, and recovery, indicating different drought levels or drought that occurred in the past.

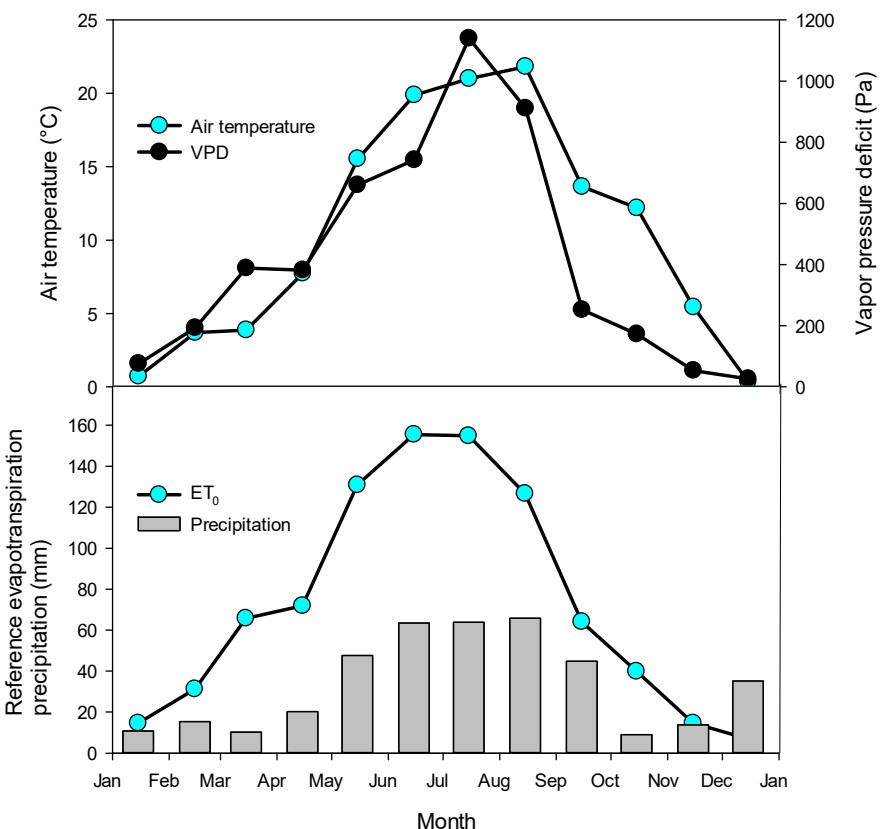

**Figure 1.** Weather at the research plot in 2022. Upper panel: monthly means of air temperature and vapor pressure deficit (*VPD*). Lower panel: monthly sums of precipitation and the reference evapotranspiration.

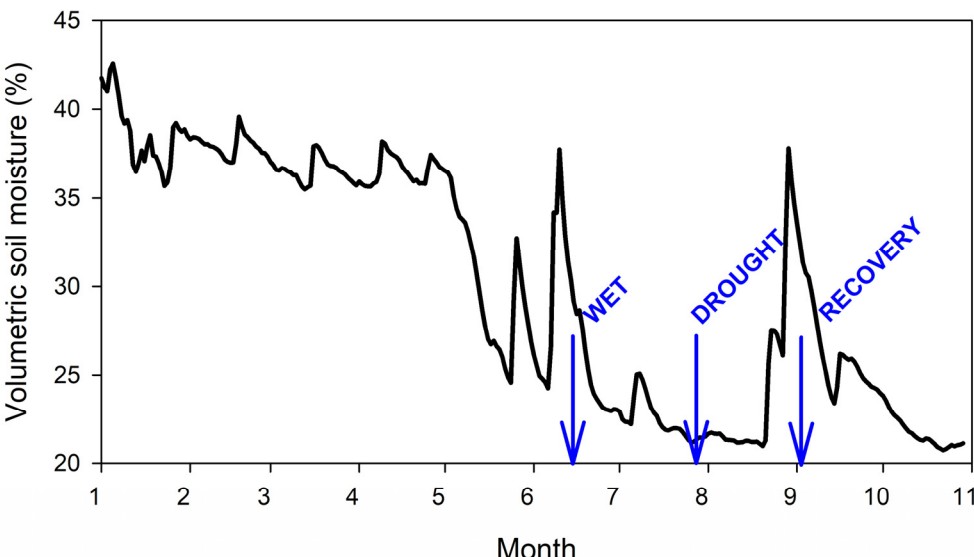

**Figure 2.** Volumetric soil moisture at the research plot. Three blue arrows indicate the days of measurement of photosynthesis.

　　　Thirteen broadleaved species (*Acer campestre, Acer platanoides, Carpinus betulus, Crataegus monogyna, Fraxinus excelsior, Ligustrum vulgare, Prunus mahaleb, Prunus spinosa, Quercus robur, Rhamnus cathartica, Sambucus nigra, Tilia cordata, Viburnum lantana*) were measured (Table 1). Two MultispeQ V 2.0 (PhotosynQ, East Lansing, MI, USA) [35] instruments were employed to obtain the following parameters: relative chlorophyll content (expressed as the index *SPAD:* special products analysis division), the quantum yield of photosystem II

($\Phi_{\text{II}}$), linear electron flow (*LEF*), the proportion of energy dissipated as non-photochemical quenching ($\Phi_{\text{NPQ}}$), and non-regulatory energy dissipation ($\Phi_{\text{NO}}$). The leaves were easily accessed due to the fact that the tree crowns were close to ground level. They remained attached to the tree during the measurements. All of the leaves were measured on the edge of the habitat corridor, and they were sun leaves. The species were measured 77–167 times/species (respectively, 12–78 times/species/day). We chose one leaf per tree. The time of the measurements on every individual day was between 8:30 a.m. and 4 p.m. (daylight saving time). In total, 1785 measurements were taken.

*2.3. Data Analysis*

Variables *SPAD*, $\Phi_{\text{II}}$, *LEF*, $\Phi_{\text{NPQ}}$, and $\Phi_{\text{NO}}$ measured by MultispeQ were calculated using the following equations set by the manufacturer:

$$SPAD = k \times relative\ chlorophyll \tag{1}$$

where *k* is the arbitrary (and proprietary) correlation coefficient used in the Minolta SPAD, but approximated using the MultispeQ V 2.0 calibration cards; and relative chlorophyll is:

$$relative\ chlorophyll = log_{10}\left(\frac{Abs_{940mm}/ref.\ Abs_{940mm}}{Abs_{650mm}/ref.\ Abs_{650mm}}\right) \tag{2}$$

$$LEF = \Phi_{\text{II}} \times PAR \times 0.4 \tag{3}$$

where *PAR* is photosynthetically active radiation measured by MultispeQ for leaf surface during every measurement, 0.4 is the coefficient of energy partitioning between photosystems I and II multiplied by the PAR absorbance by leaf and

$$\Phi_{\text{II}} = \frac{F'_m - F_s}{F'_m} \tag{4}$$

where $F'_m$ is the maximum fluorescence yield during a saturating light pulse applied to light-exposed leaves, $F_s$ represents the fluorescence emission in light-adapted samples;

$$\Phi_{\text{NO}} = \frac{F_s}{F_m} \tag{5}$$

where $F_m$ is the maximum fluorescence yield during a saturating light pulse applied to dark-adapted leaves and

$$\Phi_{\text{NPQ}} = 1 - \Phi_{\text{II}} - \Phi_{\text{NO}} \tag{6}$$

Analysis of MultispeQ data was performed in R 4.1.2 [36] through RStudio 2023.06.0 (RStudio Inc., Delaware Corporation, Boston, MA, USA) [37]. Data were sorted and curated using the "dplyr" package [38]. Differences in *SPAD* were analysed using ANOVA and then by Tukey post hoc test of multiple comparisons of the means. Values of *SPAD* lower than 20 were excluded from the analysis. The relationship of *LEF* and *PAR* was modelled by generalised nonlinear least squares fit method in the "nlme" package [39] using the following equation:

$$LEF = LEF_{max}\left[1 - \exp(-\frac{\alpha \times PAR}{LEF_{max}})\right] \tag{7}$$

where $LEF_{max}$ is the maximal value of *LEF* and $\alpha$ is the slope. Predicted values for every species at all three days of measurement (wet, drought, recovery) were plotted over the measured points using the "ggplot2" package [40]. We tested the effect of the interaction of the day of the measurement and species on the relationship of $\Phi_{\text{NPQ}}$ and $\Phi_{\text{NO}}$ to the PAR using a linear regression model. Estimated mean values and their contrasts were estimated in the "model-based" package [41] using Holm p-value adjustment method. Plots of interaction effects in linear regression models were designed using the "interactions"

package [42] and later adjusted by the "ggplot2" package to unify them with others. Weather and soil data were visualised using Sigmaplot 12.5 (Systat Software Inc., Chicago, IL, USA, 2016). The confidence level used for all analyses was 0.95 and the significance level was alpha = 0.05.

## 3. Results

### 3.1. Weather and Soil

The mean air temperature in 2022 was 10.5 °C, minimal −11.4 °C in December, and maximal 36.1 °C in August (Figure 1). The average vapor pressure deficit (VPD) was 418 Pa, reaching a maximal value of 4544 Pa in summer. The mean global radiation was 124 W m$^{-2}$. The sum of precipitation was 400 mm year$^{-1}$ when most of the rain occurred in the summer months (June–August). The sum of reference evapotranspiration was 877 mm per year, which exceeded precipitation in most of the months. The climatic water deficit thus reached 477 mm in 2022.

Soil moisture was highest in winter and spring while at its lowest in summer (July, August) which is typical for central Europe (Figure 2). Maximum volumetric soil moisture exceeded 40% in winter while minimum moisture was 21% in August. On the three days of measurement, the soil moisture was 30.3% on 14 June (wet), 21.3% on 25 July (dry), and 32.4% on 2 September (recovery).

### 3.2. Chlorophyll Content in Leaves

Different species had different concentrations of chlorophyll in their leaves as estimated by the *SPAD* index ($p < 2 \times 10^{-16}$) (Figure 3). The highest chlorophyll content was found in shrubs such as *Viburnum lantana*, *Ligustrum vulgare*, and *Sambucus nigra*. *Quercus robur* had the highest chlorophyll content from the trees, followed by *Tilia cordata* and the two species of *Acer*. The lowest chlorophyll content in leaves was observed in *Carpinus betulus* trees reaching *SPAD* 38, which was significantly lower than any other species. The second lowest amount of chlorophyll was in the leaves of *Fraxinus excelsior* which also significantly differed from all other species.

### 3.3. Chlorophyll Fluorescence

3.3.1. Linear Electron Flow

The $LEF_{max}$ during the wet season significantly differed between species (Figure 4, Supplementary Table S1). The highest *LEF* was found in *Quercus robur*. *Prunus spinosa*, *Viburnum lantana*, and *Crataegus monogyna* were three other species with above-average $LEF_{max}$. The lowest $LEF_{max}$ was in *Carpinus betulus* and *Tilia cordata*.

Drought affected the *LEF* in eight species ($p < 0.001$ in all species, Figure 4, Supplementary Table S2). The $LEF_{max}$ declined significantly only in *Tilia cordata* and *Viburnum lantana*. The coefficient indicating the slope of the regression between the *LEF* and *PAR* changed in seven out of thirteen species. The *LEF* recovered in most species after the release from drought with three exceptions: the slope of the regression remained lower in September than during June in *Acer campestre*, *Fraxinus excelsior*, and *Tilia cordata*.

3.3.2. Non-Photochemical Losses

Non-photochemical losses of energy responded to the drought. The ratio between the non-photochemical quenching ($\Phi_{NPQ}$) and non-regulatory energy dissipation ($\Phi_{NO}$) changed due to drought. The $\Phi_{NPQ}$ increased during the soil desiccation in 11 out of 13 species (Figure 5, Supplementary Table S3). The mean $\Phi_{NPQ}$ (as estimated by the linear model, Supplementary Table S3), when the soil was wet in June, ranged between 29% in *Prunus mahaleb* and 46% in *Carpinus betulus*. During drought, the $\Phi_{NPQ}$ increased in most species, except *Ligustrum vulgare* and *Sambucus nigra*. After the release from drought, in September, most of the woody species maintained the same or higher levels of $\Phi_{NPQ}$ as during the drought. The exceptions were *Tilia cordata* and *Carpinus betulus*.

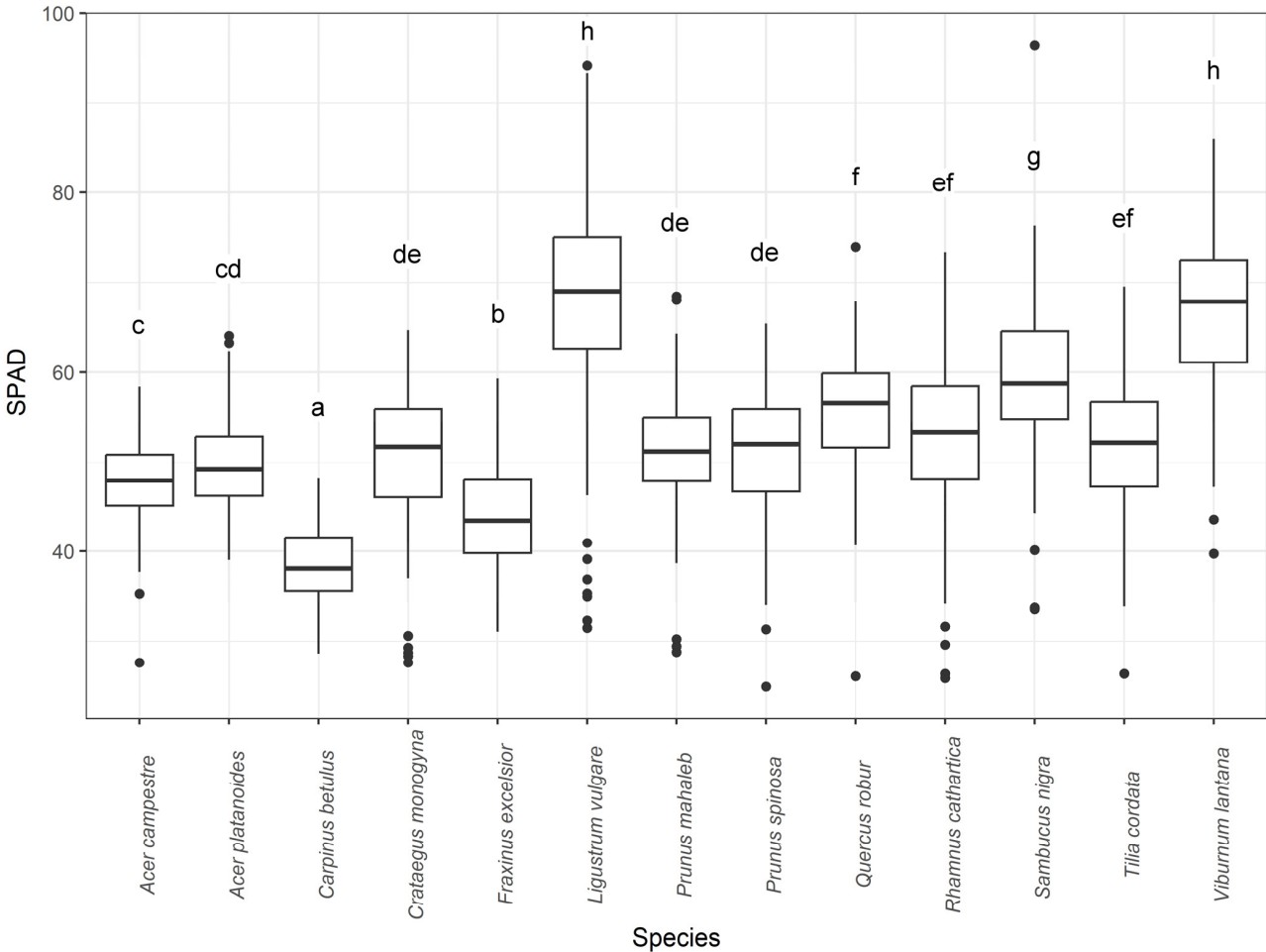

**Figure 3.** The relative concentration of chlorophyll in the leaves of trees and shrubs. Different letters indicate that the values of SPAD are significantly different. The box indicates the interquartile interval, where 50% of the data is found. The vertical line in the box indicates median. Data points outside this interval are represented as black points on the graph and considered potential outliers.

The changes in $\Phi_{NO}$ due to drought and in recovery were mostly inverse to the changes in $\Phi_{NPQ}$ (Figure 6, Supplementary Table S4). The lowest values of $\Phi_{NO}$ in wet June were found in *Carpinus betulus* and the two species of *Acer*. The highest value of 30% was in *Quercus robur*. During drought, the $\Phi_{NO}$ decreased in all species. The lowest values were found at *Carpinus betulus* (7%) and highest in *Quercus robur* (22%). After the release from drought, the $\Phi_{NO}$ remained the same in all species except *Tilia cordata*.

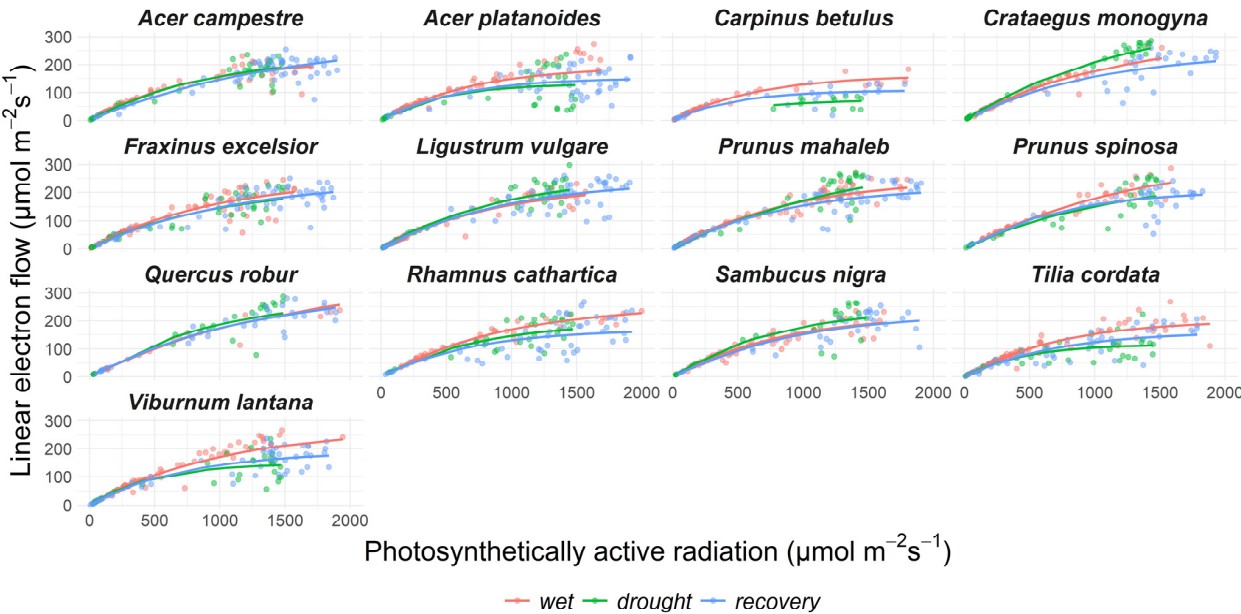

**Figure 4.** The dependence of linear electron transport in PSII on the photosynthetically active radiation. The points are showing measured values, lines are modelled according to Equation (7) above.

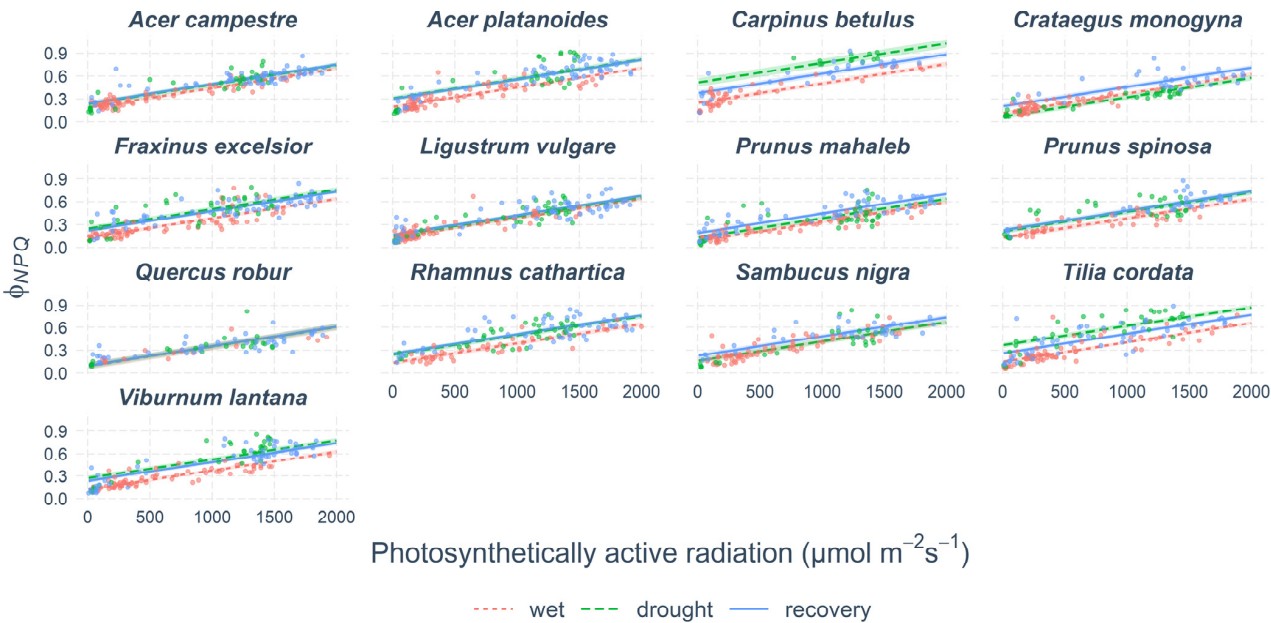

**Figure 5.** The portion of energy lost as non-photochemical quenching ($\Phi_{NPQ}$). Points show measured values. Lines show modelled predictions and coloured bands show 95% confidence intervals.

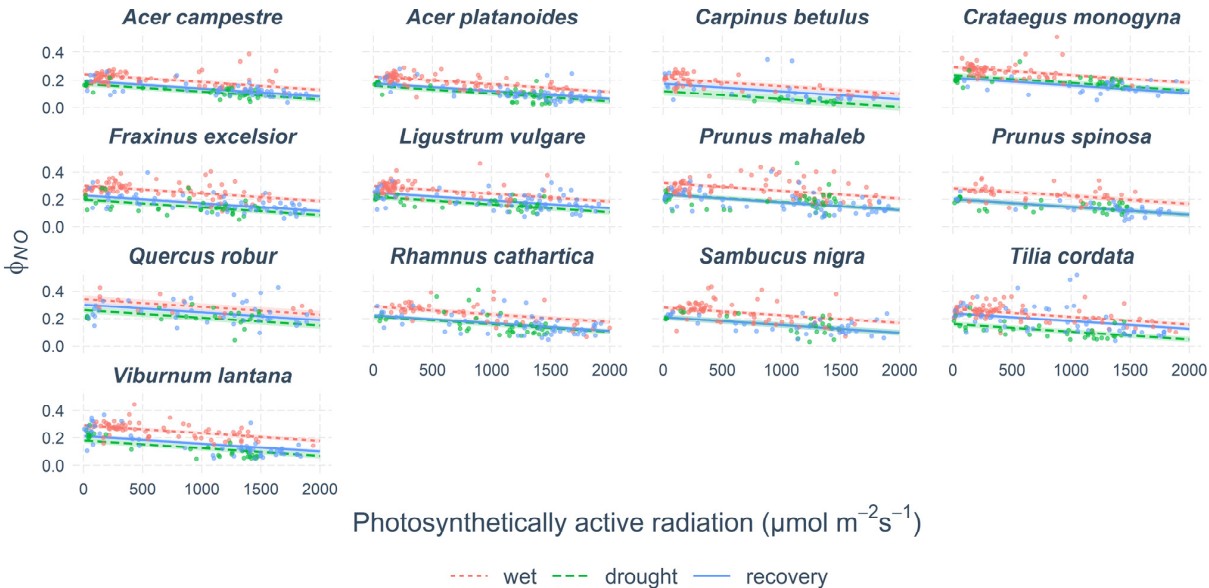

**Figure 6.** The portion of energy lost in a range of non-regulated processes ($\Phi_{NO}$). Points show measured values. Lines show modelled predictions and coloured bands show 95% confidence intervals.

## 4. Discussion

Chlorophyll fluorescence indicated how drought affected leaf-level physiology. The 13 woody species employed different strategies of adjustment of their primary processes of photosynthesis to drought and subsequent recovery. About half of the species exhibited a decline in *LEF* with its subsequent recovery in most species. A greater manifestation observed than changes in *LEF* was the shift from non-regulated dissipation of excess energy towards regulated non-photochemical quenching. While the changes in *LEF* were transient in most species and the *LEF* increased after the release from stress, the changes in the ratio between $\Phi_{NPQ}$ and $\Phi_{NO}$ were mostly permanent.

In this work, we utilised the steady-state chlorophyll fluorescence measured on sun-adapted leaves. This approach has both advantages and disadvantages in describing the photosynthesis-related physiology of leaves. The main advantage of this approach is that we measure the processes during the day when the leaves are photosynthetically active. Moreover, the leaf does not require the dark adaptation, which takes at least 15 min, but is better overnight [23,43]. On the other hand, the measurements are slightly more time-consuming than simple measurements of $F_v/F_m$. With MultispeQ, it took about 20 s to obtain the reading. The *LEF* provides information about the number of electrons transported between the photosystems. The value is directly related to actual photosynthesis and photorespiration. However, its interpretation is more complicated than a simple $F_v/F_m$ value. Common knowledge sets the value of $F_v/F_m$ in healthy leaves close to 0.8. A value lower than 0.66 indicates structural damage to leaves [44]. Therefore, $F_v/F_m$ can be used as a simple index of leaf vitality and damage across the species. However, $F_v/F_m$ is rather insensitive to the small levels of drought [30,31] and this index significantly declines only when the leaf photochemistry is severely damaged.

The slow kinetics of steady-state chlorophyll fluorescence is a better indicator of drought stress than $F_v/F_m$. For example, the yield of photosystem II and electron transport rate decreased in European beech trees due to drought, while non-photochemical losses increased [45], as found in our study. The decline of *LEF*, however, was not very pronounced and occurred only in half of the species. It is likely that the decrease in *LEF* increase in *NPQ* applies mostly to the short episodes of stress [46] and not to the long-lasting and recurrent periods of drought [47]. Long-term increases in *NPQ* may have negative effects on photosynthesis and metabolism in the leaf. Therefore, a decrease in *LEF* and an increase in *NPQ* often dissipates excess energy only during relatively short periods [48].

During drought, the gas exchange strongly declines but electron flow in the primary reactions of photosynthesis changes only mildly. The resulting ratio of gas exchange photosynthesis to electron transport rate may be used as another indicator of the relative level of drought stress and the degree of isohydry of a specific species [49]. About 10 electrons are needed for the fixation of a molecule of $CO_2$ in C3 species. During drought, this ratio increases. The reason why we did not employ a combination of gas exchange and chlorophyll fluorescence in the screening of a wide range of species is time: measurements of gas exchange are about one order of magnitude slower (and an order of magnitude more expensive) than chlorophyll fluorescence.

The ratio between energy dissipated in the means of regulated ($\Phi_{NPQ}$) and non-regulated ($\Phi_{NO}$) processes was the best indicator of drought. As the drought increased, the $\Phi_{NPQ}$ increased while the $\Phi_{NO}$ decreased in all species. The change was permanent in all species, except *Tilia cordata*. $\Phi_{NPQ}$ describes the quantum yield which is dissipated by the downregulation of photochemical processes and $\Phi_{NO}$ describes the quantum yield which is dissipated by other nonphotochemical losses [50]. Plants in stress employ *NPQ* mechanisms to protect themselves from excessive excitation energy by harmless heat dissipation ($\Phi_{NPQ}$). On the other hand, $\Phi_{NO}$ often leads to the production of harmful reactive oxygen species [51]. Trees need to find a balance between photoprotection and productivity, since too much photoprotection may decrease the growth [52]. Here, we demonstrated that the woody plants were able to shift from the $\Phi_{NO}$ to $\Phi_{NPQ}$ during summer drought. In other words, the trees were able to increase the rate of dissipation of excess energy by the xanthophyll cycle and decrease the production of harmful reactive oxygen while *LEF* remained similar.

Chlorophyll fluorescence indicated several woody species, which differed from the other trees and shrubs in their drought response. A potentially sensitive species identified in the habitat corridor was small-leaved lime (*Tilia cordata*). While its photoprotective system acclimated to drought, the acclimation did not last as long as in other species. This finding matches the finding comparing acclimation to drought between field maple and small-leaved lime [53]. The study found that field maple trees that experienced drought grew better in subsequent droughts than controls, while the drought acclimation did not help the lime trees. *Tilia cordata* also suffers more from the occurrence of drought than other trees in extreme urban environments: a single drought period results in a growth decline for the next several years [54]. The leaf level study that focused on fluorescence induction of several species planted along roads indicated that lime was very sensitive while *Acer campestre* was resistant [55], which is synonymous to the findings in this study. One of the reasons for the leaf damage may be the isohydry of lime trees [56], which results in a need for high rates of dissipation of excess energy. Still, the sensitivity of the small-leaved lime must be taken into account within the context of the extreme conditions of habitat corridors. A recent study comparing lime to the European beech found lime to be more resistant than beech: beech from our experience would not survive in the habitat corridor [57].

Oak, *Quercus robur*, was indicated by this study as another species that may potentially be damaged by drought. The reason is the highest values of $\Phi_{NO}$. A dendrochronological study comparing multiple tree species in harsh environments indicated that *Quercus robur* was indeed similarly sensitive to drought as *Tilia cordata* [58]. Mature oaks often suffer from upper-crown damage due to drought [59]. On the other hand, anisohydry allows oaks to maintain high levels of carbon assimilation, therefore utilising the high *LEF*. Still, in the long term, oak is less sensitive to drought stress than small-leaved lime, while the most resistant is another ring-porous species, ash (*Fraxinus excelsior*) [60]. The likely reason is the deep root system of ashes and oaks in comparison to the shallower roots of limes and the difference in their stomatal regulation towards *VPD* compared to diffuse-porous species [60,61].

The European hornbeam, *Carpinus betulus*, had the lowest *LEF* of all studied species. Low photosynthesis led to slower growth rates compared to other tree species in the habitat corridor. Low *LEF* may indicate drought sensitivity [26]. At the same time, the highest

$\Phi_{NPQ}$ and lowest $\Phi_{NO}$ allowed hornbeam to effectively dissipate excess energy without the production of harmful reactive oxygen species. Nevertheless, hornbeam is extremely sensitive to high temperatures [62]. Therefore, its use in habitat corridors requires shelter from faster-growing trees. The next slower-growing species was field maple (*Acer campestre*). While average in most photochemical parameters such as *LEF*, it has a very low $\Phi_{NO}$. Field maple also exhibits high tolerance to extreme temperatures [62]. These characteristics predispose field maple to grow on the sun-exposed edges of the habitat corridors [55]. Still, shrubs, such as *Crataegus monogyna*, may be even more suitable to edges, due to high light demand and even better drought tolerance [63]. Indeed, the *LEF* of shrubs such as *Prunus spinosa*, *Viburnum lantana*, and *Crataegus monogyna* was among the highest, exceeded only by pedunculate oak, indicating high drought resistance [26]. With substantially lower $\Phi_{NO}$ than in oak, these shrubs will be resistant to drought. These species are therefore suitable for the sunny edges of habitat corridors.

## 5. Conclusions

Measurements of chlorophyll fluorescence helped us to identify potentially drought-resistant and potentially drought-vulnerable woody species. The most useful indices were excess energy dissipated by the means of regulated ($\Phi_{NPQ}$) and non-regulated ($\Phi_{NO}$) processes. *Tilia cordata*, a species commonly planted in habitat corridors, was found to be potentially vulnerable to drought, due to its isohydry and inability to maintain increased rates of dissipation of the excess energy in a harmless way. Oaks' (*Quercus robur*) foliage may potentially be damaged by the excess energy, but its anisohydry and likely good access to soil water help it to maintain high growth rates. Field maple (*Acer campestre*), hornbeam (*Carpinus betulus*), and shrubs such as *Prunus spinosa*, *Viburnum lantana*, and *Crataegus monogyna* were indicated as drought-resistant woody species, suitable for planting at the edges of habitat corridors.

**Supplementary Materials:** The following supporting information can be downloaded at: https://www.mdpi.com/article/10.3390/f14081521/s1, Figure S1: Global radiation and air temperature during the days of measurement of physiological parameters. Wet day in June (upper panel), Dry day in July (central panel) and Recovery in September (lowest panel). Data are in one hour time resolution; Figure S2: Drought index developed by Intersucho for 24 July 2022 (i.e. closest available date to the day of dry measurements). White cross at the southeast part of Czechia indicates the locality of the habitat corridor. Dark red color indicates exceptionally strong to extreme drought; Table S1: Coefficients of maximum linear electron flow ($LEF_{max}$) and initial slope of the regression between PAR and *LEF*; Table S2: Coefficients of maximum linear electron flow ($LEF_{max}$, here denoted as 'a') and initial slope of the regression between PAR and *LEF* (here denoted as 'b'). The values for the wet variant are coefficients for the wet conditions in June. The coefficients for drought and recovery need to be added to or subtracted from the coefficient for drought. The *p*-value indicates, whether is the coefficient for drought or recovery different from wet conditions in June; Table S3: $\Phi_{NPQ}$ during wet, drought and recovery (upper part of the table). Comparison between wet, drought and recovery for each of the species (bottom); Table S4: $\Phi_{NO}$ during wet, drought and recovery (upper part of the table). Comparison between wet, drought and recovery for each of the species (bottom).

**Author Contributions:** Conceptualisation, J.U., M.M. and L.Ú.; methodology, M.M., J.U., B.J. and L.Ú.; software, M.M.; validation, M.M. and J.U.; formal analysis, M.M. and J.U.; investigation, J.U., M.M., B.J. and L.Ú.; resources, B.J. and L.Ú.; data curation, M.M.; writing—original draft preparation, J.U.; writing—review and editing, M.M., B.J., W.R. and L.Ú.; visualisation, M.M. and J.U.; project administration, L.Ú. and B.J.; funding acquisition, L.Ú. and B.J. All authors have read and agreed to the published version of the manuscript.

**Funding:** This research was funded by the Technological Agency of the Czech Republic, Programme Environment for Life, and is an output of project SS01010174 Functionality of the Territorial System of Ecological Stability and its Perspective in the Context of Global Climate Change and by project No. 21-11487S of the Czech Science Foundation (Adaptation, avoidance, or extinction: linking community ecology and ecophysiology to understand the moisture deficit effects in temperate forests).

**Data Availability Statement:** The data that support the findings of this study are available online on the PhotosynQ webpage (https://www.photosynq.org/projects/uses-1/explore and https://www.photosynq.org/projects/uses-2/explore (both accessed 20 July 2023).

**Acknowledgments:** The authors wish to thank Yan Zvyniatskovskyi and Ludmila Habánová for their help with field measurements and Daniel Volařík for his statistical advice.

**Conflicts of Interest:** The authors declare no conflict of interest.

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
