# Peer review of "Effect of Drought on Photosynthesis of Trees and Shrubs in Habitat Corridors"

_forests, doi:10.3390/f14081521_

Round 1

Reviewer 1 Report

1. All figures except figure 1 needs significant improvement not only in picture resolution but also think about how to present this data.

2. How can the authors parse out that only draught stress shows these observations?

3. Why does one species shows so level of SPAD? Do you have any explanations to this?

4. Please proofread manuscripts to fix grammatical errors and typos.

5. It would really help if you can simply or concise the discussion section.

N/A

Author Response

We thank the reviewer for the time spent on our paper and for the comments that helped us to improve the manuscript. We reflected all comments in the revised text of the manuscript.

Reviewers’ comment: 1. All figures except figure 1 needs significant improvement not only in picture resolution but also think about how to present this data.

Authors’ answer: We provided high-resolution Figures alongside the Figures that were embedded in the Word document. We believe, that even though the automated conversion of Figures made in R and pasted in the Word document may decrease their resolution below an acceptable level, Forests makes sure to use only the high-resolution Figures during publishing. We also improved the presentation of data in all Figures except No. 1.

Reviewers’ comment: 2. How can the authors parse out that only draught stress shows these observations?

Authors’ answer: As in any field study, no factor can be perfectly controlled. Since we relied on a naturally occurring drought and measured a large number of fully grown trees, we did not have any fully watered control. This is the limitation of the study that must be taken into account when interpreting the results.

Reviewers’ comment: 3. Why does one species shows so level of SPAD? Do you have any explanations to this?

Authors’ answer: The hornbeam, Carpinus betulus, had the lowest levels of SPAD. It’s SPAD well matched the lowest values of LEF. Hornbeam is a tree species typically occurring in the understory of the forest, where it is well-adapted to low-light conditions. This may be the evolutionary reason for low concentrations of chlorophyl in the hornbeam leaves.

Reviewers’ comment: 4. Please proofread manuscripts to fix grammatical errors and typos.

Authors’ answer: The manuscript was edited for English language by native speaker William Robb. In spite of the editing, some typos still remained in the text. We removed them during the revision. If there are still some remaining typos which we missed, we would appreciate highlighting them in the PDF and we will fix them before publishing.

Reviewers’ comment: 5. It would really help if you can simply or concise the discussion section.

Authors’ answer: We shortened the discussion section as suggested.

Reviewer 2 Report

This study focused on the assessment of plant drought resistance of 13 tree and shrub species commonly used in habitat corridors in the Czech Republic by quantifying their chlorophyll fluorescence characteristics before, during, and after the 2020 summer drought. I found the topic of this study interesting but less reported, and the results could provide important insights into tree performance in those ecological importance habitat corridors in response to increasing drought in a warming climate. And the study seems well-designed and conducted. But I do have some concerns regarding methodology and data analysis.

 1) I do not quite follow the data analysis, e.g., linear/nonlinear regression between chlorophyll fluorescence parameters and PAR, shown in the manuscript. Why conduct these regression-type analyses between plant performance and PAR, what is the logic behind this? Why not just simply compare the means of LEF, PhiNPQ, and PhiNP of a certain species under wet/dry/recovery conditions to evaluate plant drought resistance? Which in my opinion is much clearer and more straightforward.

 2) the details regarding the chlorophyll fluorescence measurements, the methodology part, should be improved. Information needs to be added including a) exact measurement time as photosynthesis has diurnal dynamics and how to control the confounding effects of photosynthetic diurnal changes on your measurements; b) sample size? in other words, how many individuals and leaves were measured per species per day? c) from which part of the canopy did you select and measure fully developed leaves? and did you measure the leaves destructively? or in situ by using a ladder (?) to reach the top canopy?

 3) it seems that the three sampling days (‘wet’, ‘dry’, and ‘recovery) are rather arbitrarily chosen. At least from what is shown in fig1, I do not see why the ‘dry/drought’ day should be considered a drought event. And what are the justifications for selecting those three days for measurements? You will need to show long-term climatic patterns to justify that the days you chose are indeed wet, dry (a drought event), and recovery days. Additionally, plants' responses to drought/soil moisture are greatly confounded by factors like plant age, plant phenology, antecedent soil moisture, and so on. I am not quite convinced by a single-day measurement ('dry day') conducted here to robustly conclude plant drought resistance.

 4) please improve the quality of all figures, and for results showing in supplement information, please make tables according to the journal guidelines/requirements.

 Please also see other comments embedded in the PDF document. 

English is generally fine.

Round 2

Reviewer 2 Report

I appreciate your revision. All my comments and suggestions have been well incorporated/addressed.